# Understanding the Role of External Facilitation to Drive Quality Improvement for Stroke Care in Hospitals

**DOI:** 10.3390/healthcare9091095

**Published:** 2021-08-25

**Authors:** Tharshanah Thayabaranathan, Nadine E. Andrew, Rohan Grimley, Enna Stroil-Salama, Brenda Grabsch, Kelvin Hill, Greg Cadigan, Tara Purvis, Sandy Middleton, Monique F. Kilkenny, Dominique A. Cadilhac

**Affiliations:** 1School of Clinical Sciences at Monash Health, Monash University, Clayton, VIC 3168, Australia; nadine.andrew@monash.edu (N.E.A.); r.grimley@griffith.edu.au (R.G.); tara.purvis@monash.edu (T.P.); Monique.Kilkenny@monash.edu (M.F.K.); dominique.cadilhac@monash.edu (D.A.C.); 2Peninsula Clinical School, Central Clinical School, Monash University, Frankston, VIC 3199, Australia; 3Queensland State-Wide Stroke Clinical Network, Brisbane, QLD 4000, Australia; Greg.Cadigan@health.qld.gov.au; 4Sunshine Coast Clinical School, Griffith University, Birtinya, QLD 4575, Australia; 5Metro South Research, Metro South Health, Brisbane, QLD 4102, Australia; Enna.Stroil-Salama@health.qld.gov.au; 6Stroke Division, The Florey Institute of Neuroscience and Mental Health, Heidelberg, VIC 3052, Australia; brenda.grabsch@gmail.com; 7Stroke Foundation, Melbourne, VIC 3000, Australia; KHill@strokefoundation.org.au; 8Nursing Research Institute, St Vincent’s Health Network Sydney, St Vincent’s Hospital Melbourne, Australia and Australian Catholic University, Sydney, NSW 2010, Australia; Sandy.Middleton@acu.edu.au

**Keywords:** quality improvement, stroke, facilitation, behavior change intervention, process evaluation, improvement science

## Abstract

The use of external facilitation within the context of multicomponent quality improvement interventions (mQI) is growing. We aimed to evaluate the influence of external facilitation for improving the quality of acute stroke care. Clinicians from hospitals participating in mQI (Queensland, Australia) as part of the Stroke123 study were supported by external facilitators in a single, on-site workshop to review hospital performance against eight clinical processes of care (PoCs) collected in the Australian Stroke Clinical Registry (AuSCR) and develop an action plan. Remote support (i.e., telephone/email) after the workshop was provided. As part of a process evaluation for Stroke123, we recorded the number and mode of contacts between clinicians and facilitators; type of support provided; and frequency of self-directed, hospital-level stroke registry data reviews. Analysis: We measured the association between amount/type of external facilitation, (i) development of action plans, and (ii) adherence to PoCs before and after the intervention using AuSCR data from 2010 to 2015. In total, 14/19 hospitals developed an action plan. There was no significant difference in amount or type of external facilitator support provided between hospitals that did, and did not, develop an action plan. There was no relationship between the amount of external facilitation and change in adherence to PoCs. Most (95%) hospitals accessed stroke registry performance data. In the Stroke123 study, the amount or type of external facilitation did not influence action plan development, and the amount of support did not influence the changes achieved in adherence to PoCs. Remote support may not add value for mQI.

## 1. Introduction

Despite widely available evidence supporting clinical interventions that improve health outcomes for patients hospitalized for acute stroke, adherence to these recommended interventions is often suboptimal [1,2]. In Australia, there is evidence that adherence to recommended clinical processes of care (PoCs) varies between hospitals that treat patients with acute stroke [3]. Factors that may contribute to this variation include the ability to keep up to date with current evidence, clinicians’ values and attitudes, willingness to change practice, and time constraints [4,5]. Such discrepancies are concerning given that variability in acute patient care has been shown to adversely affect patient outcomes, and it results in ineffective use of health care resources [6].

Grimshaw et al. provided an overview of the effectiveness of professional behavior change strategies available to support quality improvement (QI) activities at a hospital level [7]. These strategies have been categorized by the Cochrane Effective Practice and Organization of Care Group [8] and include techniques such as educational outreach visits; audit and feedback; interprofessional collaboration; reminders; and tailored interventions. However, the most effective implementation strategies may be those that utilize a multicomponent approach [5,9]. The authors of a more recent review offered no compelling evidence on what is an effective strategy to change clinician behavior in acute care settings [10]. Therefore, understanding what components of a QI intervention are effective in changing clinicians’ behaviors to reflect best evidence is challenging.

Although many factors may influence the translation of evidence into practice, active facilitation in some form is often required to elicit change in behavior and, thus, practice [4]. Facilitation is simply a technique by which one person makes things easier for others by aligning the available evidence to the particular context [11]. This may be provided by either internal (i.e., inside the local hospital) and/or external (i.e., outside of the local hospital) facilitators [12]. Facilitation is also a key element of The PARiHS framework (integrated—Promoting Action on Research Implementation in Health Services) that has proved to be a useful practical and conceptual tool for many researchers and clinicians in quality improvement programs [11]. Facilitators are individuals with the specific roles, skills, and knowledge to help clinicians, teams, and organizations deliver evidence-based care by prompting change in their professional behavior [13]. They have a key role in helping individuals and teams understand what, and how, they need to change practice. Some key roles of external facilitators are the following: identify, engage, and connect stakeholders; facilitate collaboration including the development of action plans; support communication and information sharing; provide updates on evidence-based strategies; and evaluate practice change efforts [14]. However, very little is known about the type or amount of external facilitation that is required to achieve changes in practice.

The Stroke123 study was a pragmatic, real-world before and after registry-based study designed to assess the effectiveness of financial incentives (implemented to improve access to stroke units) coupled with a multicomponent quality improvement (mQI) intervention [15]. Collectively, the financial incentives program and the mQI were the intervention tested in the Stroke123 study. The mQI, known as the Enhanced StrokeLink program, included audit and feedback processes, education and action plan development through a single onsite workshop provided by an external facilitator with ongoing remote support (telephone/email) after the workshop (see Online Appendix A). Overall, there was an 18% improvement in the change in median composite score (i.e., summarized in a single measure as the proportion of all needed care that was given) for adherence to <=8 PoCs measured in the Australian Stroke Clinical Registry (AuSCR) across the study periods (95% CI, 12–24%). In secondary outcome analyses, the Enhanced StrokeLink program was associated with a non-statistically significant additive increase in the primary composite score (4%), and a 3% additive increase in composite score limited to the PoCs included in action plans across study periods [15]. In Stroke123, not all hospitals developed action plans during the workshop. We are also unsure of whether the additional support provided outside of the workshops increased the success of this mQI and sought to explore these issues further using data obtained as part of the process evaluation for Stroke123.

The objective of this study was to provide process evaluation information in relation to the external facilitator role as part of the mQI component of the Stroke123 intervention. Specifically, we sought to evaluate (i) the influence of external facilitation on whether or not action plans were used for improving the quality of acute stroke care as part of the mQI, and (ii) to assess the association between amount/type of external facilitation and adherence to the PoCs specified within the action plans before and after the mQI.

## 2. Materials and Methods

### 2.1. Study Design

This paper features a process evaluation for the mQI intervention (Enhanced StrokeLink program) of a pre–post observational study with audit and feedback.

### 2.2. Population

Staff from public hospitals providing acute stroke care located in the state of Queensland, Australia that collect data using the AuSCR [16] were eligible to participate in the mQI intervention (*N* = 23). The program was delivered between March 2014 and November 2014. Ethics approval was obtained from all participating hospitals, with the lead ethics committee in Queensland being the Metro South Human Research Ethics Committee (HREC/13/QPAH/31). This study was carried out in accordance with the Declaration of Helsinki. The AuSCR has ethics approval to use an opt-out method of consent for collection of patient clinical data.

### 2.3. QI Intervention Description

A detailed description of the overall Stroke123 study and data sources has been previously reported [15,16]. In a previous publication, the Alberta Context Tool has been used to describe aspects of organizational context that have affected the delivery of care in these hospitals prior to the workshops being delivered by the facilitator. This preliminary research revealed important insights about the role of context, including that culture and social capital were the main aspects of organizational context affecting the delivery of evidence-based care rather than differences in perceptions of context based on clinician background or the location of the stroke service (e.g., metropolitan or rural location) [17].

The mQI workshop component was based on the Plan–Do–Study–Act (PDSA) method [18], where feedback was provided to clinicians on their hospital performance, and action plans were developed to improve adherence to clinical guideline recommendations. The workshop conducted once at each hospital was run by an external facilitator employed by the Stroke Foundation who had a clinical background and a range of skills including quality improvement methodology [15,19]. As part of the audit and feedback processes, the AuSCR data were used to (1) identify practice gaps, (2) focus the development and implementation of action plans for areas the local staff at these hospitals wanted to work on, and (3) provide a reliable method to routinely monitor performance since the PoC data are collected on consecutive admissions (i.e., adherence to acute stroke PoCs). At the face-to-face workshop (scheduled for ≈3 h duration) the external facilitators educated the participating hospital staff on the current best practice recommendations associated with the PoCs collected in AuSCR. They also highlighted local gaps in practice using the AuSCR data, identified barriers to adherence to these PoCs, and provided customized, site-specific strategies as suggestions to address identified barriers to improve practice. Time permitting, the hospital staff then discussed and agreed on a focused action plan that was to address the PoCs selected to work on improving [16]. Additional support was also provided by external facilitators from the Queensland State-wide Stroke Clinical Network (i.e., leads) and the AuSCR (i.e., project coordinator). Queensland State-wide Stroke Clinical Network clinical leads provided education on a range of topics, and they also promoted networking and shared learning with other stroke clinicians throughout the state. AuSCR project coordinators assisted in the extraction and interpretation of the PoC data, at the time of performance gap identification, action plan design, and implementation. All external facilitators provided ongoing peer support when required, via telephone, face-to-face, and/or email after the initial workshop.

### 2.4. Data Collection

In order to capture the amount and mode of facilitator support for the Stroke123 process evaluation, a specifically designed data collection tool was developed to enable the prospective and standardized collection of activity data from the external facilitators. Items recorded included the frequency, duration, and mechanism of external facilitation episodes (Table 1). The tool included pre-specified categories to ensure standardized data capture.

The amount of external facilitation was determined by the frequency and duration of professional behavior change support provided to clinicians, mode of support delivery, and time spent delivering support (including the workshop). These parameters were used to understand the influence of external facilitation required to achieve an effect from action plans for this current study.

### 2.5. Clinical Indicator Data to Assess Quality Improvement Effects

As part of developing and implementing action plans, there were eight acute stroke PoCs collected in AuSCR within Queensland available as options for quality improvement. These options included: treatment in a stroke unit; in acute ischemic stroke, use of intravenous thrombolysis, aspirin <48 h, and prescription of antiplatelet/other antithrombotic medication at discharge; early patient mobilization; use of a swallow screen/assessment before feeding; prescription of antihypertensive medication at discharge; and use of a discharge care plan if the hospital is separate to the community [15].

AuSCR data (from 2010 to 2015) were used as a proxy to assess the successful implementation of action plans by comparing changes in the mean composite score (see below) before and after the development of action plans as part of the mQI program.

### 2.6. Calculation of Composite Score

The composite score is calculated as the proportion of patients with the documented measure (i.e., acute stroke PoCs) divided by the number of patients eligible for the measure. Where applicable, eligibility criteria for a PoC are specified and decision rules around eligibility are the same as the Stroke123 results paper [15]. For example, adherence to the administration of aspirin within 48 h is relevant only for patients presenting with ischemic stroke or transient ischemic attack. The percent adherence was calculated for each PoC for the participating hospitals. Details in how the composite score was calculated are provided in the Stroke123 results paper [15]. For this study, the absolute change in composite scores between the pre-intervention phase and post-intervention phases was calculated and stratified by whether hospitals developed an action plan and by the amount and mode of facilitator support.

### 2.7. Data Analyses

Descriptive analyses were undertaken for the discrete and continuous data used for this study. Accurate data for the time duration for writing an email was infeasible to obtain, so we assumed that each email would take an average time of 15 min. Normality tests were conducted to check the distribution of each continuous variable. The Wilcoxon rank-sum test was conducted to explore the association between the development of an action plan or not and the amount of external facilitation provided for each hospital during the intervention period. Mean and standard deviation were calculated for Hedge’s g, which is an effect size used to indicate the standardized difference between two means when sample sizes are <20. Hedge’s g statistics of 0.20 to 0.49 are considered small standardized effect sizes, 0.50 to 0.79 are considered medium standardized effect sizes, and 0.80 or greater is considered a large standardized effect size [20].

A *p*-value of <0.05 was considered statistically significant. Data were analyzed using Stata (Stata Statistical Software, Release 12.0; Stata Corporation, College Station, TX, USA).

## 3. Results

Staff from 19 of 23 eligible Queensland hospitals agreed to participate in the mQI intervention (urban hospitals: 55%; rural hospitals: 45%; 16% were teaching hospitals). During the intervention delivery phase (March 2014 to November 2014), nine different external facilitators provided support to the 19 hospitals. The most frequent activities provided by the facilitators were educational outreach (42%), which was followed by interprofessional collaboration (30%), review of audit data (11%), and reminders (6%) (Figure 1).

During the workshops, 14/19 (74%) hospitals developed action plans. Five out of 19 hospitals did not develop an action plan following the workshop: the reasons included already having developed a QI plan and local staff changes. Among the 14 hospitals that developed an action plan, most hospitals selected eight processes of care to work on. Throughout the intervention, 18/19 (95%) hospitals accessed their AuSCR data for performance monitoring (overall number of times accessed AuSCR online data reports: median (Q1, Q3): 12 [7,18]).

The number of times external facilitators provided support and the mode of contact during the intervention period is shown in Table 2. There was a clinically relevant but not statistically greater amount of external facilitator support in hospitals that developed an action plan when compared to hospitals that did not develop an action plan. This finding was relevant overall as well as for the subgroups of mode of contact (Table 2, *p* = 0.308). Based on Hedge’s g, the effect size of external facilitation provided by face-to-face only was considered medium; telephone was small; and the effect size for total contact time via all modes was medium.

Among the 9/14 hospitals that developed an action plan, the adherence to the PoCs, as measured by mean composite scores, improved post-intervention (Figure 2 and Figure 3). Moreover, 4/5 hospitals that did not develop an action plan also observed an increase in adherence to these PoCs as determined by the mean composite scores (Figure 2).

There was no relationship between amount of external facilitation and absolute change in composite scores (Figure 4 and Figure 5). However, in some hospitals, it appears that support provided by external facilitators to staff at participating hospitals may have had a positive impact on adherence to the acute stroke processes of care.

## 4. Discussion

This is one of the few studies in which the influence of external facilitators has been assessed for stroke based on the PDSA model. In this process evaluation study, we had a specific focus on the mQI component of the Stroke123 intervention (financial incentives plus mQI) that was previously assessed as part of a controlled before and after study, and it was found to be highly effective [15]. In the current study, we found that on average, seven hours more contact time with staff was provided by facilitators for hospitals that developed an action plan. We did not find a statistically significant association between the amount or mode of external facilitation and the development or success of the action plans with improving process of care adherence. All hospitals improved their performance relative to historical performance, and those with action plans appeared to do better with the receipt of more facilitator time. Perhaps external facilitation cannot be viewed as an individual component, but it is one among many integral components of a PDSA model and our PARiHS framework-based mQI intervention that has contributed to improving adherence to these acute stroke PoCs, as demonstrated in the Stroke123 study [15]. Since the access to financial incentives as part of the Stroke123 intervention was available to all hospitals, it is unlikely to have differentially impacted our assessment of the association between external facilitation and action planning presented in this study.

Few mQI studies include process evaluations with details of the amount and mode of facilitation provided. It is important to evaluate the processes of a study so that other researchers can reliably implement or build on the current intervention. Based on the Template for Intervention Description and Replication (TIDieR) checklist, some of the key features should include reporting of duration, dose or intensity, and mode of contact, as these can influence the effectiveness and potential replicability of the program [21]. All hospitals received a single face-to-face workshop with standardized components. However, the final element, the action plan, was not always delivered due to time constraints. This provided an opportunity to assess whether action plans make a difference in addition to the amount and mode of support provided by external facilitators within the context of acute hospital care settings. The results, based on Hedge’s g, suggests that although support from external facilitators may be useful, much is unknown about the ideal amount or mode of facilitation required to support change. Since all hospitals received a workshop, it appears that additional remote support may not add value for this type of mQI.

To some extent, our results may reflect reverse causation whereby these correlations may have been due to the following: (1) not targeting an achievable or manageable number of processes of care may have led to inefficient efforts and greater facilitator time; and (2) hospitals with significant barriers to improvement, or less effective internal facilitation, may have had a greater need for external assistance whilst having a reduced ability to impact change. It is possible that external facilitation is necessary, but the amount of external facilitation provided may depend on the number of issues needed to be addressed, staff availabilities, organizational context, and resources available per hospital [22]. Increasing amounts of external facilitation may not necessarily result in improved outcomes. It is also important to note the complexity of strategies chosen for improvement. Some changes take longer to implement than others and, in some circumstances, although a specific strategy may be implemented, it might take a longer period to see change in adherence as measured by the AuSCR data.

Several different modes of delivery were provided by different external facilitators. The presence of an external facilitator who provides support (via face-to-face and telephone) may ensure action plans are developed and implementation is followed through to support change in practice. Quality-of-care activities in the ‘Get With The Guidelines Stroke’ program evaluated the processes of workshops and impact of facilitation, which were shown to improve adherence to all primary performance measures, i.e., improve quality of care [23].

In this study, we noted that throughout the mQI, staff from most hospitals accessed the AuSCR data consistently for performance monitoring. This observation indicates that data from the AuSCR are important and can be used by staff to assess their hospital’s performance and make relevant changes where necessary as part of their routine QI practice. A limitation is that we were unable to directly quantify access to the data reports available for hospital staff to download at any time from the AuSCR and include this information in our analysis.

The strengths of this study are the number and variety of hospitals that participated (e.g., teaching, urban and rural hospitals), offering a broad cross-section of hospitals that provide stroke care similar to other parts of Australia or other countries. Additionally, a standardized data collection tool was used to collect data from all external facilitators. All external facilitators had a nursing or allied health background and completed an extensive standardized training program designed and delivered by the Stroke Foundation in order to prepare them for their role [15]. Limitations of this study include the inability of five hospitals to develop an action plan as part of their face-to-face workshop. Evidence from complementary research suggests that future approaches to PDSA model-based studies and action planning may need to be undertaken in more than one workshop or stages. For example, a two-step process may be beneficial whereby the external facilitators initially educate and identify the local barriers and enablers at the hospital, followed by a second meeting that is completely focused on action planning [24]. Another limitation is that contact data were self-reported and only reflected the facilitators’ perspectives, and it did not include those of the hospital staff receiving the support. In addition, no information on internal facilitation at a hospital level, if any, was available. Although most of the time, the external facilitators provided data on a fortnightly basis, on some occasions, there was a two-month delay. Hence, there may have been potential for recall bias. Nevertheless, the external facilitators kept personal diaries and email records, which provide objective data to ensure that any recall bias was minimized. In-depth data on the content of the professional behavior change support provided were not available. Our research presented here offers a case study based on the context of stroke, which may or may not be applicable to other settings. Nevertheless, our findings highlight important insights into how much facilitator time may be required to assist clinicians, teams, or organizations to address variation in evidence-based care by prompting change in their professional behavior based on Plan–Do–Study–Act methods relevant to their local hospital setting. Our study provides useful data to guide the design of future research, especially with respect to the investment in external facilitators. In future studies, there is a need to evaluate the effectiveness of external facilitation from those receiving in-depth support (i.e., clinicians and hospital staff) to directly assess the impact on adherence to care processes and patient outcomes. With the sample size of 19 hospitals, our study was potentially underpowered to detect statistically significant differences for the primary aim of this study. Our results provide much food for thought and assist to advance the field.

## 5. Conclusions

We were unable to demonstrate a significant relationship between amount or mode of contact of external facilitation and the development of action plans or change in adherence to acute stroke processes of care. External facilitation cannot be viewed as an individual component but may be an integral component of a complex mQI intervention. Further work is required to understand the optimal amount and mode of contact of external facilitation to enable efficient design of mQI interventions. It may be that additional remote support may not add value to this type of mQI.

## Figures and Tables

**Figure 1 healthcare-09-01095-f001:**
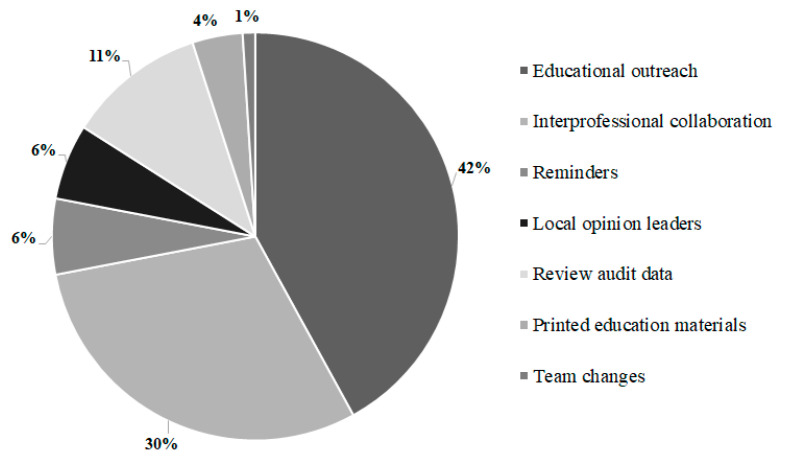
Proportion of professional behavior change interventions (categorized by the Cochrane Effective Practice and Organization of Care Group) [8] provided by the external facilitators.

**Figure 2 healthcare-09-01095-f002:**
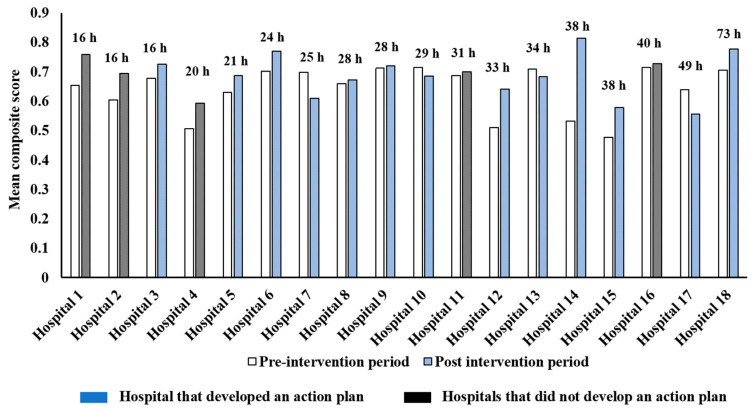
Change in mean composite score from <=8 acute stroke processes of care collected in all QLD hospitals participating in the multicomponent quality improvement program—Enhanced StrokeLink. Support provided by external facilitators to the respective hospitals during the multicomponent quality improvement program—Enhanced StrokeLink period is displayed in hours (i.e., h). Missing AuSCR data for one hospital.

**Figure 3 healthcare-09-01095-f003:**
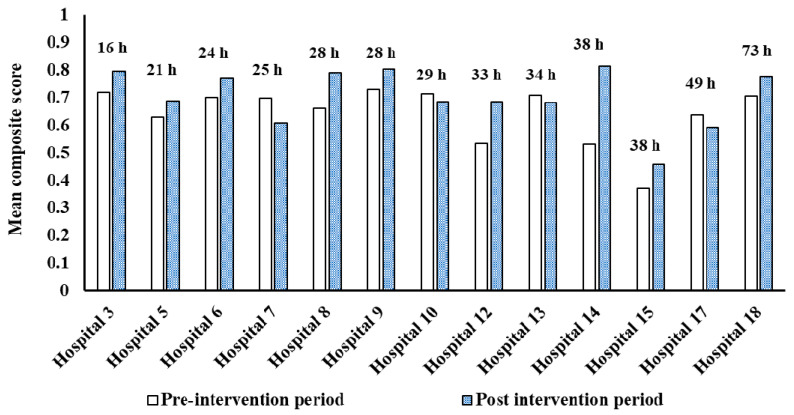
Change in composite score from acute stroke processes of care nominated by hospitals in their action plans (i.e., only includes hospitals that developed an action plan, and acute stroke processes of care nominated by hospitals in their action plan).

**Figure 4 healthcare-09-01095-f004:**
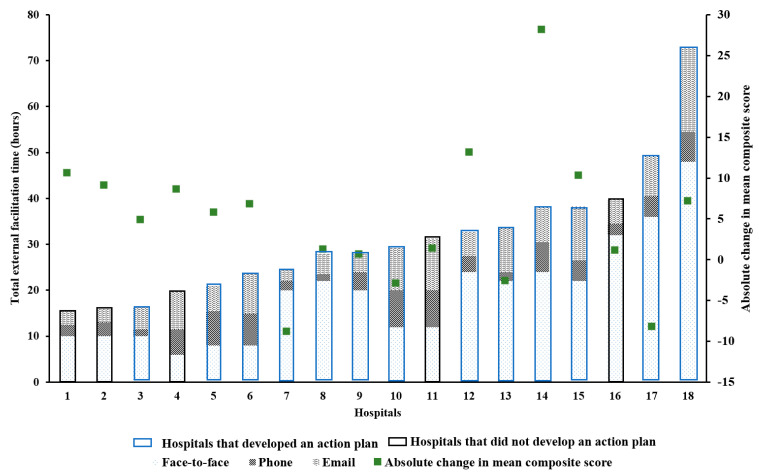
Absolute change in composite score from <=8 acute stroke processes of care collected in all QLD hospitals vs. support provided by external facilitators to the respective hospitals during the multicomponent quality improvement program—Enhanced StrokeLink period is displayed in hours. Missing AuSCR data for one hospital.

**Figure 5 healthcare-09-01095-f005:**
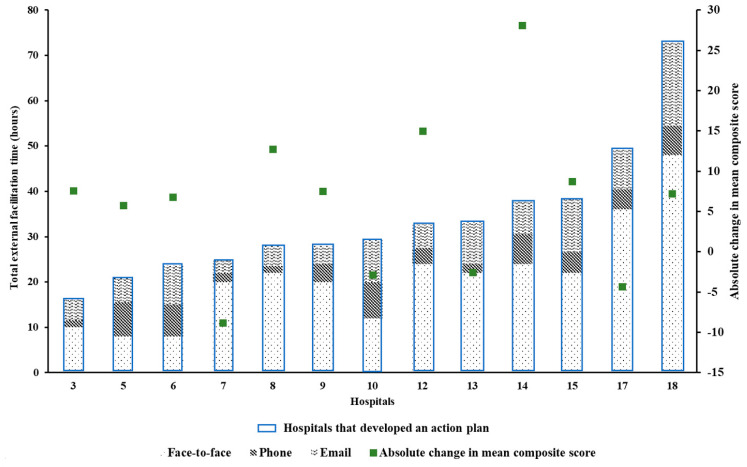
Absolute change in composite score from acute stroke processes of care nominated by hospitals in their action plans vs. support provided by external facilitators to the respective hospitals during the multicomponent quality improvement program—Enhanced StrokeLink period is displayed in hours.

**Table 1 healthcare-09-01095-t001:** Information collected from external facilitators.

Type of Information
Number of contacts the external facilitators had with the relevant clinicians and hospital staff
Mode of contact (e.g., telephone, face-to-face, email)
Contact time in minutes for telephone and face-to-face contacts
Professional behavior change support type provided (e.g., reminders, educational outreach) *
Action plan initiated by staff at participating hospitals (Yes/No)
Hospitals accessing their AuSCR data (by downloading online live reports)—Yes or No

* defined by the classifications by the Cochrane Effective Practice and Organization of Care Group [8].

**Table 2 healthcare-09-01095-t002:** Median and mean hours of external facilitator support and development of action plans.

Mode of Contact	Measure	All Hospitals *N* = 19	Hospitals that Developed an Action Plan *N* = 14	Hospitals That Did Not Develop an Action Plan *N* = 5	*p* Value	Hedge’s g
Face-to-face	Median (Q1, Q3)	20 (10, 24)	21 (10, 24)	10 (10, 12)	0.243	N/A
	Mean (± SD)	19 (11)	20 (11)	14 (10)	0.300	0.557
Telephone	Median (Q1, Q3)	5 (3, 7)	5 (3, 7)	3 (3, 6)	0.676	N/A
	Mean (± SD)	5 (2)	5 (2)	4 (3)	0.411	0.440
Email	Median (Q1, Q3)	6 (4, 9)	7 (4, 9)	5 (5, 8)	0.817	N/A
	Mean (± SD)	7 (4)	7 (4)	7 (3)	0.859	0
Total	Median (Q1, Q3)	22 (15, 29)	29 (24, 38)	20 (16, 31)	0.308	N/A
	Mean (± SD)	30 (14)	32 (15)	25 (10)	0.350	0.501

## Data Availability

The datasets used and/or analyzed during the current study are available from the corresponding author on reasonable request.

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
