# Peer review of "Understanding the Role of External Facilitation to Drive Quality Improvement for Stroke Care in Hospitals"

_healthcare, 2021, doi:10.3390/healthcare9091095_

Round 1

Reviewer 1 Report

I read with great interest the manuscript about the role of external facilitation in quality improvement for stroke process. The authors have analyzed the data from the Stroke123 intervention, with economic incentives and a multimodal quality improvement intervention, consisting of 1 workshop in each of the different hospitals and posterior facilitator assistance in different modes (face-to-face, telephone and mail). They were not able to demonstrate that the amount of external facilitation or subtype could influence the quality improvement.

I think it is an interesting manuscript that open questions instead of answering them. My only concern is the reproducibility of the results of the study in other settings. Different hospital organization, cultures, and even people believes would probably receive the external facilitation differently. Therefore, in my opinion, generalization and usefulness of the study is questionable.

Reviewer 2 Report

This report describes the association between external facilitation on hospital staff and the subsequent improvement in their stroke care as measured by the adherence to the selected standard cares. This intervention, a component of the “multicomponent” intervention constituting the Stroke 123 study, itself consisted of multiple components, and the investigators attempted to associate each component to selected outcomes. It is usually difficult to attribute the effect of any “multicomponent” interventions, whatever they may be—quality improvement, patient safety, or infection prevention, to their individual components, the difficulty that also arose in this study, which was not able to any significant findings. Nevertheless, I believe that the descriptive presentation in this manuscript could give readers some insights, thereby helping them to their own hypothesis to work with, so that publication of this report is meaningful. My comments are as follows.

  1. I cannot fully understand what the concrete intervention was; the description of the intervention is abstract. I would like to see, possibly as a supplemental material, a hand-out or video clip of on-site work-shop, examples of action plans, reminders, educational outreach, follow-up contacts, and so forth.
  2. What was the rationale for employing the development of action plan as a primary outcome? This outcome, among the potential outcomes to be employed in this study, appears the farthest from patient outcome. To me, the composite score appears closer, and patient outcome measures such as 90-day mRS much closer. Indeed, figures 2 and 4 does not seem to show any action-plan-dependent improvement in composite scores. 
  3. Did you evaluate or control for “facilitator effect?” Intuitively, the effect of intervention would depend on the facilitators’ skill.
  4. As a part of the Stroke123, the apparent improvement by the facilitation may be due to carry-over of preceding financial incentives. This is a potential limitation, which should be discussed.
  5. As the authors mention, the potential reverse causation is a major concern of this study; because substantial part of the external facilitation seems to have been given as needed, the amount of facilitation may reflect low-quality care provided by the corresponding hospitals. Because such needs can be manifested as baseline composite scores, I would suggest showing both before-and-after scores, not just showing absolute difference.
  6. Line 303–307: Did you find any association between the frequency of hospitals’ access to the AuSCR and the improvement in the scores?
